# Purification of Colon Carcinoma Cells from Primary Colon Tumor Using a Filtration Method via Porous Polymeric Filters

**DOI:** 10.3390/polym13193411

**Published:** 2021-10-05

**Authors:** Jia-Hua Wang, Lee-Kiat Ban, Henry Hsin-Chung Lee, Yen-Hung Chen, Hui-Yu Lin, Zhe-Wei Zhu, Her-Young Su, Akihiro Umezawa, Abdulrahman I. Almansour, Natarajan Arumugam, Raju Suresh Kumar, Gwo-Jang Wu, Akon Higuchi

**Affiliations:** 1Department of Chemical and Materials Engineering, National Central University, Taoyuan 320, Taiwan; a0922416954@gmail.com (J.-H.W.); henrychen880409@gmail.com (Y.-H.C.); linhuiyu0613@gmail.com (H.-Y.L.); slkdfert@gmail.com (Z.-W.Z.); 2Department of Surgery, Hsinchu Cathay General Hospital, Hsinchu 300, Taiwan; cgh06321@cgh.org.tw (L.-K.B.); hsinchuoff@cgh.org.tw (H.H.-C.L.); 3Graduate Institute of Translational and Interdisciplinary Medicine, National Central University, Taoyuan 320, Taiwan; 4Department of Obstetrics and Gynecology, Bobson Yuho Women and Children’s Clinic, Hsinchu 302, Taiwan; yuhoclinic@gmail.com; 5Department of Reproduction, National Center for Child Health and Development, Tokyo 157-8535, Japan; umezawa-a@ncchd.go.jp; 6Department of Chemistry, College of Sciences, King Saud University, Riyadh 11451, Saudi Arabia; almansor@ksu.edu.sa (A.I.A.); anatarajan@ksu.edu.sa (N.A.); sraju@ksu.edu.sa (R.S.K.); 7Graduate Institute of Medical Sciences and Department of Obstetrics & Gynecology, Tri-Service General Hospital, National Defense Medical Center, Taipei 114, Taiwan; 8Department of Chemical Engineering and R&D Center for Membrane Technology, Chung Yuan Christian University, Taoyuan 320, Taiwan; 9Nano Medical Engineering Laboratory, Riken Cluster for Pioneering Research, Riken, Saitama 351-0198, Japan

**Keywords:** poly(lactide-*co*-glycolic acid), silk, filter, colon carcinoma cell, cancer stem-like cell, filtration, carcinoembryonic antigen, colony-forming unit

## Abstract

Cancer stem cells (CSCs) or cancer-initiating cells (CICs) are key factors for tumor generation and metastasis. We investigated a filtration method to enhance CSCs (CICs) from colon carcinoma HT-29 cells and primary colon carcinoma cells derived from patient colon tumors using poly(lactide-*co*-glycolic acid)/silk screen (PLGA/SK) filters. The colon carcinoma cell solutions were permeated via porous filters to obtain a permeation solution. Then, the cell cultivation media were permeated via the filters to obtain the recovered solution, where the colon carcinoma cells that adhered to the filters were washed off into the recovered solution. Subsequently, the filters were incubated in the culture media to obtain the migrated cells via the filters. Colon carcinoma HT-29 cells with high tumorigenicity, which might be CSCs (CICs), were enhanced in the cells in the recovered solution and in the migrated cells based on the CSC (CIC) marker expression, colony-forming unit assay, and carcinoembryonic antigen (CEA) production. Although primary colon carcinoma cells isolated from colon tumor tissues contained fibroblast-like cells, the primary colon carcinoma cells were purified from fibroblast-like cells by filtration through PLGA/SK filters, indicating that the filtration method is effective in purifying primary colon carcinoma cells.

## 1. Introduction

Tumors have a small subpopulation of cancer stem cells (CSCs) or cancer-initiating cells (CICs) that have self-renewing potential and are considered to be the origin of tumors in patients [1,2,3,4]. CSCs (CICs) are considered to be a key initiator of tumor creation, metastasis, and proliferation [5,6,7,8,9,10,11]. Several CSC markers have been reported, such as Lgr5, Musashi-1, ALDH-1, CD24, CD166, CD44, and CD133, although it is debatable whether these markers are real CSC or CIC markers. It seems that CSC (CIC) markers depend on the primary tissue, timing, and researchers who are investigating CSCs (CICs). It is difficult to isolate CSCs (CICs) utilizing isolation procedures such as fluorescence-activated cell sorting (FACS) or magnetic-activated cell sorting, which target CSC (CIC) markers. Tumor generation by transplanting cells and colony-forming unit assays is typically performed to assess tumor cell line establishment and for CSC (CIC) identification.

Colon carcinoma, which is also known as bowel cancer or colorectal cancer, is cancer developed from the colon or rectum and is one of the most fatal and common gastrointestinal cancers in the world [12]. The purification and isolation of tumorigenic colon CSCs (CICs) from primary colon tumor tissues should be valuable for the development of novel diagnostic and personal therapeutic treatments in the future.

In our previous study [13], we intended to purify CSCs (CICs) from the colon carcinoma cell line, LoVo cells (epithelial human cells), utilizing the filtration method via poly(lactide-*co*-glycolic acid)/silk screen (PLGA/SK) filters with a pore size of (*r*) = 20–30 μm and nylon mesh (NM) filters with *r* = 11 and 20 μm. Colon carcinoma (LoVo) cell line solutions were permeated via filters to obtain a permeation solution. Then, Dulbecco’s modified Eagle’s medium (DMEM) was permeated via the filters to obtain the recovered solution, in which the cells adhered to the filters were released into the recovered solution. Subsequently, the filters were cultivated in the cell cultivation media to obtain the migrated cells from the filters. The cells migrated via the filters showed higher generation of colony formation and expressed higher CSC (CIC) surface markers of CD133 and CD44 than that of the LoVo cells cultured on tissue culture polystyrene (TCP) dishes (control). The migrated cells from the filters also generated more CEA (carcinoembryonic antigen, colon cancer marker protein) than that of the control LoVo cells on TCP dishes. However, this research only targeted one specific cell line of colon cancer cells. The isolation of CSCs (CICs) from primary colon carcinoma cells has not yet been performed using the filtration method.

In this study, we investigated the purification of CSCs (CICs) of another colon carcinoma cell line, HT-29, as well as primary colon carcinoma cells from colon tumor tissues of patients utilizing the filtration method via NM and PLGA/SK filters. HT-29 was selected to be used in this study because HT-29 cells are one of the most common colon cancer cell lines. There was no morphological difference between HT-29 cells and other colon cancer cell lines such as Colo205 and LoVo cells. The isolation efficiency was characterized using (i) CD44 and CD133 marker expression, (ii) colony-forming unit assay, and (iii) CEA production. We expected to establish a purification method for a primary colon carcinoma cell line with a high proportion of colon CSCs (CICs) from the patient’s tumor tissue using the filtration method.

## 2. Materials and Methods

### 2.1. Ethical Statement

All experiments in this research were approved by the ethics committees of Cathay General Hospital (CGH-P108082). Every experiment was processed in accordance with all applicable and relevant governmental and institutional regulations and guidelines. We obtained informed consent from the patients.

### 2.2. Materials

The biomolecules, biomaterials, and chemicals utilized in this research are summarized in Table 1. The other materials utilized in this study were obtained from Sigma–Aldrich (St. Louis, MO, USA).

### 2.3. Formation of PLGA/SK Filters

PLGA/SK filters were prepared using a freeze-extraction process (Figure 1A), which was reported previously [14,15,16,17,18,19]. Briefly, 10, 5, and 3 wt.% PLGA solutions were prepared using PLGA with *N,N*-dimethylformamide as a solvent [20,21,22,23,24,25]. Three sheets of silk screen mesh (170 mesh size, 3.3 cm diameter) were inserted into glass Petri dishes (diameter = 3.5 cm), which were pretreated with high-vacuum grease to avoid filter adhesion. Subsequently, PLGA solution (3 mL) was added into the dishes containing the silk screen mesh and placed in a refrigerator at −20 °C for 24 h. The frozen PLGA/SK solution was placed in 74–76% ethanol at −21 °C for 120 h, and the 74–76% ethanol was replaced twice a day to remove the solvent (*N,N*-dimethylformamide) and to generate a porous configuration (PLGA/SK filters). Then, the PLGA/SK filters were placed in a fume hood at 22–24 °C for 48 h to remove residual ethanol (Figure 1A). PLGA/SK filters generated with 10%, 5%, and 3% PLGA solution were named after the PLGA-10/SK, PLGA-5/SK, and PLGA-3/SK filters, respectively. UV light irradiation was used to sterilize the filters on a clean bench for one day before the PLGA/SK filters were used via the filtration method.

### 2.4. Characterization of Filters

The surface and/or cross-sectional morphology of the PLGA/SK and NM filters was evaluated by scanning electron microscopy (SEM, S-3000H, Hitachi, Tokyo, Japan). Surface 3D images of the PLGA/SK filters were also characterized using a PRIMOS surface analyzer (PRIMOS Lite, GFM, Berlin, Germany).

### 2.5. Cultivation of Colon Carcinoma Cell Lines

The colon carcinoma cell line HT-29 was obtained from the Food Industry Research and Development Institute (BCRC 60148; Hsinchu, Taiwan). HT-29 cells were cultivated in DMEM supplemented with 10% fetal bovine serum (FBS) and a 1% antibiotic–antimycotic solution at 37 °C in a 5% CO_2_ incubator and passaged utilizing the conventional protocol [1].

### 2.6. Isolation of Primary Colon Carcinoma Cells from Patient Colon Tumor Tissues

Adenocarcinoma colon carcinoma tissue from a patient (approximately 1 cm × 1 cm × 1 cm size, female, 56 years old) in a tube containing saline water with 1% antibiotic–antimycotic from the surgery room was quickly transported onto a clean bench. The isolation process of primary colon carcinoma cells from patient colon tumor tissues is described schematically in Figure 1B. Colon carcinoma tissue was removed from the tube and immersed in a 12-well TCP dish with PBS containing 1–5% antibiotic–antimycotic to rinse the colon carcinoma tissue. The colon carcinoma tissue was rinsed with fresh PBS containing 1–5% antibiotic–antimycotic for 30 s 12 times. Subsequently, the colon carcinoma tissue was placed into a 10 cm TCP dish and minced into small sections that were no more than 0.1–1 mm^3^ with a sterile scalpel blade. Then, 2 mL collagenase type IV (100 U/mL) and 1 mL CaCl_2_ solution (3 mM) were added to the minced colon carcinoma tissue. The minced colon carcinoma tissue solution was inserted into a 50 mL centrifuge tube and incubated in a water bath at 37 °C with vigorous shaking for 1 h. Collagenase digestion was performed to breakdown the large tissue fragments into small fragments. After the collagenase digestion process, an equal volume of cell culture medium (DMEM) was added to the tube to neutralize the digestion reaction, and the solution was permeated through a filter with *r* = 100 μm to remove large and undigested tissue fragments. Subsequently, the permeated solution was centrifuged at 400× *g* for 420 s. Subsequently, the supernatant was removed, and the cell pellets were resuspended in 2 mL of ACK lysis buffer to lyse red blood cells. After the cell solution was incubated for 5 min, 2 mL of DMEM was added to the cell solution to neutralize the reaction, and the cell solution was centrifuged at 400× *g* for 420 s. Then, the supernatant was removed, and the cell solution was resuspended in 2 mL of PBS. The solution was centrifuged again at 400× *g* for 420 s to remove the reaction agent as much as possible. Finally, 1 mL of cell culture medium (DMEM with 10% FBS and 1% antibiotic–antimycotic) was added to the cell solution and agitated gently to make a homogeneous primary colon carcinoma cell solution. The primary colon carcinoma cell solution was seeded on TCP dishes and cultivated in a 5% CO_2_ incubator for 168 days. The media were exchanged every 2–10 days depending on the attached cell number. Cells were passaged when they reached approximately 80% confluence.

### 2.7. Purification of CSCs (CICs) by the Filtration Method

The human colon carcinoma (HT-29) cell solution or primary colon carcinoma cell solution was filtered through NM filters with *r* = 11 µm (NM-11) or PGLA-3/SK, PGLA-5/SK, and PGLA-10/SK filters using a batch-type membrane holder (C161, Millipore Corp.) as described previously for blood cell [26,27] or adipose-derived stem cell permeation [16,28,29,30,31]. Figure 1C shows a schematic representation of a filtration protocol to purify the CICs (CSCs) of colon carcinoma cells from a colon carcinoma cell line (HT-29) and primary colon carcinoma cells.

The colon carcinoma cell solution (1 × 10^6^ cells in 10 mL) was filtered via NM or PLGA/SK filters at 24–26 °C with a filtration speed of 1 mL/min. The colon carcinoma cell numbers in the permeation and feed solutions (*X*_p_ and *X*_f_, respectively) were analyzed by flow cytometry (Figure 1C). The filter holder was inverted after the cell solution filtration, and DMEM containing 10% FBS (the recovered solution) was permeated via the filters at a filtration speed of 1 mL/min at 25 °C. This process was performed to obtain the cells that had adsorbed on the filters in the recovered solution (Figure 1C). The number of colon carcinoma cells in the recovered solution (*Xr*) was also analyzed by flow cytometry.

The permeation rate was evaluated from Equation (1):Permeation rate (%) = (*X*_p_/*X*_f_) × 100(1)

The recovery rate was analyzed from Equation (2):Recovery rate (%) = (*X*_r_/*X*_f_) × 100(2)

The residual rate was estimated from the following calculation:Residual rate (%) = 100 (%) − (Permeation rate (%) + Recovery rate (%))(3)

The filters were taken from the membrane holder and placed into TCP cell culture dishes containing DMEM with 10% FBS supplement after filtration of the recovered solution (washing solution). The residual cells adsorbed on the filters started to migrate away from the filters into the TCP cell culture dishes during culture for 6 days at 37 °C in a 5% CO_2_ incubator. The migrated cell numbers were evaluated utilizing flow cytometry.

The CSC marker (CD133 and CD44) expression on colon carcinoma cells in the recovered solution, permeation solution, and feed and on migrated cells was analyzed using flow cytometry after staining with antibodies against CD133 or CD44 and 7-AAD (for live and dead staining).

### 2.8. Colony-Forming Unit Analysis

Colony-forming unit analysis is a typical method to characterize the in vitro anchorage-independent cell growth ability, which can be utilized to identify CSCs (CICs) in vitro [32,33,34]. The experimental process was performed using published methods [33,34,35] with the following modifications: (1) The colon carcinoma cells were replated onto a six-well dish (10,000 cells/well) immediately after the membrane filtration method was used as the treatment. (2) For the following days, the plates were placed in the incubator until the cells formed sufficiently large colonies (approximately 9 days). (3) When the cell colonies were large enough, the cell culture was suspended. Then, the culture medium was removed from the cells, and the cells were washed with PBS three times. Two milliliters of a mixture of 6.0% (vol./vol.) glutaraldehyde and 0.5% (wt./vol.) crystal violet in H_2_O was added to the six-well dish and incubated for at least 30 min. The colonies were successfully fixed and stained during this step. (4) The mixture solution was removed from the dish and rinsed carefully with tap water. The dish with colonies was dried at room temperature. (5) The colonies were counted using a stereomicroscope.

### 2.9. CEA Production Analysis

CEA production by colon carcinoma cells was investigated utilizing an ELISA (enzyme-linked immunosorbent assay) kit [36] after the cells were filtered via NM and PLGA/SK filters. The colon carcinoma cells in the feed, recovered solution, and permeation solution as well as the migrated cells were cultured on TCP plates for 6 days, and the culture medium was changed with fresh culture medium on day 2 and 4. The culture medium used for the cell culture was taken and centrifuged at 1000× *g* for 20 min to remove cell debris. Then, the supernatant was collected after centrifugation at 400× *g* for 360 s. The concentration in the supernatant was analyzed for the CEA production rate per cell at Day 6, where the CEA production rate was evaluated from the following formula:CEA production rate (ng/10^6^ cells) = *C* × *W* × 10^6^/*Y*(4)
where *C* is the CEA concentration in the supernatant generated from colon carcinoma cells, *W* is the volume of the media (0.2 mL), and *Y* is the colon carcinoma cell number.

### 2.10. Statistical Assay

All of the quantitative data were taken from four samples. The data are shown as the mean ± standard deviation. Statistical assays were performed with an unpaired Student’s *t*-test using Excel (Microsoft Corporation). Probability values (*p*) less than 0.05 were regarded as statistically significant.

## 3. Results

### 3.1. Filter Characterization

We used commercially available nylon mesh (NM) filters with *r* = 11 μm (NM-11) and homemade poly(lactide-co-glycolic acid)/silk screen (PLGA/SK) filters made from 10%, 5%, and 3% PLGA solutions, which were named the PLGA-10/SK, PLGA-5/SK, and PLGA-3/SK filters, respectively, for isolating CICs (CSCs). These filters were characterized utilizing scanning electron microscopy (SEM). Figure 2Ai–Aiv shows outline photos of the PLGA-10/SK, PLGA-5/SK, PLGA-3/SK, and NM-11 filters. The porous structures of the filters from the photos can be expected because of the nontransparent appearance of the filters. Figure 2Av–Aviii shows the top view morphologies of the PLGA-10/SK, PLGA-5/SK, PLGA-3/SK, and NM-11 filters analyzed by SEM. NM-11 filters displayed the mesh style of regular configuration, whereas PLGA-10/SK, PLGA-5/SK, and PLGA-3/SK filters show entangled fiber configurations indicating the formation of highly tortuous pores, which were generated by microphase separation of PLGA. Figure 2Aix–Axi illustrates the cross-sections of the PLGA-10/SK, PLGA-5/SK, and PLGA-3/SK filters analyzed by SEM. The silk mesh, which serves as the backbone of PLGA/SK filters contributing to the mechanical stability of the filters, can be found in the middle of the cross-section of PLGA/SK filters. We inputted three layers of silk screen mesh in PLGA solution to generate PLGA/SK filters in this study, whereas only one layer of silk screen mesh was inserted for the preparation of PLGA/SK filters in the previous study [13,16]. This is because we wanted to both avoid pinholes in the filters and enhance the mechanical stability of PLGA/SK filters by inserting three layers of silk screen mesh. The average pore size on the filters was analyzed from their top view of SEM images utilizing ImageJ software (https://imagej.nih.gov/ij/download.html accessed on 3 October 2021) and is listed in Table 2.

The average pore size of NM-11 was 11.2 ± 2.1 μm, whereas the pore sizes of the PLGA-10/SK, PLGA-5/SK, and PLGA-3/SK filters were 28.9 ± 8.9, 34.0 ± 8.2, and 36.3 ± 12.5 μm, respectively. The pore size of PLGA/SK filters was found to decrease with the increase of PLGA concentration on the preparation of PLGA/SK filters, which was the similar tendency of PLGA/SK filters prepared using one layer of silk screen mesh in the previous study [13]. The pore size of PLGA/SK filters prepared in a previous study [13] was reported to be 22.9–28.5 μm, which depended on the PLGA concentration for the preparation of PLGA/SK filters. These pore sizes of PLGA/SK filters prepared with one layer of silk screen in a previous study [13] were found to be slightly less than the pore sizes of PLGA/SK filters prepared with three layers of silk screen in this study. Because the size of colon carcinoma cells is approximately 8–12 μm, PLGA/SK and NM filters with pore sizes similar to those of colon cells are expected to be the optimal filters for the isolation of CICs (CSCs) in the following experiments.

The surface roughness of 3D images of PLGA/SK filters was also investigated using a PRIMOS Lite surface analyzer, and 3D images of PLGA/SK filters are shown in Figure 2B. The PLGA-10/SK, PLGA-5/SK, and PLGA-3/SK filters showed extremely rough and tortuous surfaces compared to the surface of the NM-11 filters.

### 3.2. Permeation of Colon Carcinoma HT-29 Cells by the Filtration Method via PLGA/SK and NM Filters

Cells of the colon carcinoma cell line HT-29 were filtered by the filtration method via the PLGA/SK (PLGA-10/SK, PLGA-5/SK, and PLGA-3/SK) and NM-11 filters before the isolation experiments of CICs (CSCs) was performed from primary colon carcinoma cells derived from a patient’s colon tumor tissue. The morphologies of HT-29 cells in the permeation solution and recovered solution as well as the migrated cells from the filters were observed under phase-inverted microscopy, where the migrated cells were obtained from the escaped cells from the filters after cultivation of the filters for 6 days, where the colon carcinoma cells were permeated to obtain the permeation solution in advance, and subsequently the culture medium was permeated to obtain the recovered solution. The cell morphologies are shown in Figure 3A. The spherical morphology of colon carcinoma cells was observed in the recovered solution, permeation solution, and migrated cells. Furthermore, no significant difference in cell morphologies was found in the recovered solution, permeation solution, and migrated cells. Colon carcinoma cells could permeate through any filter into the permeation solution and were recovered from the solution in this study.

The residual rate, recovery rate, and permeation rate through the filters were analyzed using Equations (1)–(3) from the number of HT-29 cells in the permeation solution and recovered solutions, which were obtained using flow cytometry, and the results are shown in Figure 3B. The permeation rate through the NM-11 and PLGA/SK filters analyzed in this study was approximately 80% via any filter except the PLGA-10/SK filters (approximately 65%), which were prepared from the highest concentration of PLGA among the PLGA/SK filters. The recovery rate was much less than the permeation rate and was found to be 7–13% for any of the NM and PLGA/SK filters investigated in this study. The recovery rate via PLGA/SK filters increases with decreasing PLGA concentration in the preparation of PLGA/SK filters. Furthermore, the residual rate increased with increasing PLGA concentration in the preparation of the PLGA/SK filters. The NM-11 filters have a simpler pore structure than that of the PLGA/SK filters and showed the lowest residual rate in this study, which indicates that most colon carcinoma cells can permeate through the NM-11 filters after the detachment of colon carcinoma cells because of the simple morphology of the pore structure. It is expected that the tortuosity and complexity of the pore structure are enhanced, and the pore size decreases in PLGA/SK filters when the filters are prepared with a higher concentration of PLGA, which contributes to the enhancement of the residual rate with the increase in PLGA concentration during the preparation of PLGA/SK filters.

### 3.3. CSC Marker Expression of Colon Carcinoma HT-29 Cells after Cell Filtration via the NM and PLGA/SK Filters by the Filtration Method

Evaluating CSC (CIC) marker expression on the cells is an important index where CSCs (CICs) are purified or reduced. Therefore, the expression of CD44 and CD133 on colon carcinoma (HT-29) cells was analyzed using flow cytometry after the permeation of the cells through NM-11 and PLGA/SK filters, where colon carcinoma cells expressing CD133 [37,38,39,40] and CD44 [38,39] are typically considered CSCs (CICs).

Figure 4 shows the flow cytometry diagrams (Figure 4A (CD44) and Figure 4B (CD133)) and expression (Figure 4C (CD44) and Figure 4D (CD133)) of CSC markers (CD44 and CD133, respectively) on HT-29 cells in the permeation solution and recovered solution as well as on the migrated cells after the filtration method via NM-11 and PLGA/SK filters, where the migrated cells were obtained after cultivation of the filters in the cultivation media for 6 days. The colon carcinoma cells in the permeation solution and recovered solution as well as migrated cells expressed high CD44 and CD133. HT-29 cells cultivated on TCP dishes, which were not filtered via the NM-11 or PLGA/SK filters, expressed approximately 70.2% CD44 and 46.1% CD133. CD133 and CD44 expression of HT-29 cells cultivated on TCP dishes was much lower than that of colon carcinoma cells in the permeation solution and recovered solution as well as migrated cells after permeation with NM-11 and PLGA/SK filters (*p* < 0.05). These results indicated that colon carcinoma cells with CSC markers preferably permeate through any filters investigated in this study or that CSC marker expression on colon carcinoma cells might be enhanced after mechanical stimulation of colon carcinoma cells during permeation via the filters.

### 3.4. Colony-Forming Unit Analysis of the Colon Carcinoma HT-29 Cells Filtered via the NM and PLGA/SK Filters by the Filtration Method

Colony-forming unit assays are typically used for the evaluation of tumorigenic cells because the colony-forming ability of cancer cells is extensively related to cell tumorigenicity [32,41,42,43]. Therefore, colony-forming unit analysis of colon carcinoma (HT-29) cells was performed before and after filtration via PLGA/SK filters. Figure 5A illustrates colony-forming photos on the dishes by colon carcinoma HT-29 cells in the permeation solution (Figure 5Aa) and recovered solution (Figure 5Ab) as well as migrated cells (Figure 5Ac) after filtration through the PLGA-10/SK filters as an example. ImageJ software (https://imagej.nih.gov/ij/download.html accessed on 3 October 2021) was utilized to count the number of colonies under each condition, and the results are shown in Figure 5B.

The colony-forming numbers of colon carcinoma HT-29 cells in the recovered solution and the migrated cells after permeation via PLGA-5/SK filters were found to be higher than those on colon carcinoma HT-29 cells before the permeation of the filters, which were cultured on TCP dishes (*p* < 0.05).

### 3.5. CEA Production by Colon Carcinoma HT-29 Cells Filtered via the NM-11 and PLGA/SK Filters by the Filtration Protocol

CEA secretion by colon carcinoma cells is one of the characteristic properties of colon carcinoma cells. Therefore, CEA secretion by colon carcinoma HT-29 cells was analyzed in the permeation solution and recovered solution as well as the migrated cells after filtration via the PLGA/SK filters, and the results are shown in Figure 5C, where the CEA generated by the colon carcinoma HT-29 cells was evaluated from the concentration of CEA in the culture medium. CEA secretion was characterized from the total cell number and CEA amount produced by the cells utilizing Equation (4).

Colon carcinoma HT-29 cells in the permeation solution through PLGA-3/SK and PLGA-10/SK filters secreted a similar amount of CEA as the cells cultivated on TCP dishes (control), whereas the cells in the recovered solution and the migrated cells via NM-11 and PLGA/SK filters generated higher CEA production than the cells cultivated on TCS plates (control) (*p* < 0.05). In particular, the cells migrated via PLGA-10/SK filters secreted the highest amounts of CEA compared with other cells in the permeation solution and recovered solution via any PLGA/SK filters, as well as the migrated cells from PLGA-5/SK and PLGA-3/SK filters and extensively higher production than the colon carcinoma HT-29 cells incubated on TCP plates (*p* < 0.05).

It is suggested that colon carcinoma HT-29 cells with high tumorigenicity, which may be CSCs, could be enriched in the cells in the recovered solution and migrate through the PLGA/SK filters based on the CSC marker expression, colony-forming unit assay, and CEA production in this study.

### 3.6. Establishment of the Primary Colon Carcinoma Cell Lines from Colon Tumors

Primary colon carcinoma cells were established from the patient’s colon tumor tissues by cells mincing, collagenase digestion, and cultivation on TCP dishes following the experimental procedures described in the Methods section. We succeeded in establishing only one primary colon carcinoma cell line (M-24) from 32 trials of colon tumor tissues in this study. The primary colon carcinoma cells established in this study were characterized using CSC (CIC) marker expression and colony-forming unit assay, and the results are shown in Figure 6. CD44 and CD133 expression in primary colon carcinoma (M-24) cells is shown in Figure 6A,B, respectively, together with CD44 and CD133 expression in LoVo cells and HT-29 cells. Primary colon carcinoma cells (M-24) expressed relatively less CD44 than commercially available colon carcinoma cell lines, LoVo cells and HT-29 cells, although primary colon carcinoma cells (M-24) surely expressed the CSC (CIC) marker CD44. Interestingly, the primary colon carcinoma (M-24) cells expressed similar expression of CD133 to that of HT-29 cells. Therefore, primary colon carcinoma (M-24) cells expressed the CSC (CIC) markers CD44 and CD133 in this study.

Colony-forming unit analysis of primary colon carcinoma (M-24) cells was performed in which 10,000 cells were seeded on the dishes, and the results are shown in Figure 6C. Primary colon carcinoma (M-24) cells could successfully generate colonies, although the number of colonies formed was lower than that of commercially available colon carcinoma cells (LoVo cells and HT-29 cells). Therefore, based on CSC (CIC) marker expression and colony-forming ability, it is suggested that primary colon carcinoma cells were successfully isolated from colon tumor tissues in this study.

### 3.7. Purification of CSCs from Primary Colon Carcinoma Cells Established from Patient Colon Tumor Tissues by a Filtration Method

The primary colon carcinoma (M-24) cells established in this study contained fewer CSCs (CICs) than that of commercially available LoVo cells and HT-29 cells based on the surface marker expression of CSCs (CICs) and colony-forming ability, as discussed in Figure 6. Therefore, we tried to enhance CSCs (CICs) in primary colon carcinoma cells using the filtration method in this research. Primary colon carcinoma (M-24) cells were permeated through NM-11 or PLGA-10/SK filters, and the cells in permeation solution were obtained. Subsequently, the cultivation medium was permeated via the filters, and the cells in recovered solution were obtained. The morphologies of (a) the primary colon carcinoma (M-24) cells cultured on TCP dishes before permeation through the filters, (b) the primary colon carcinoma (M-24) cells in permeation solution and recovered solution after permeation through the filters, and (c) the migrated cells from the filters were investigated and are shown in Figure 7A.

The primary colon carcinoma (M-24) cells cultured on TCP dishes contained spherical colon carcinoma cells and spindle fibroblast cells, which indicated that the primary colon carcinoma (M-24) cells were not pure colon carcinoma cells but contained some fibroblast-like cells. After the primary colon carcinoma cells were permeated through NM-11 or PLGA-10/SK filters, most of the migrated cells showed coagulated spherical cells after permeation through both NM-11 and PLGA-10/SK filters, which are typical morphologies of colon carcinoma cells. Although fibroblast-like spindle-shaped cells could be observed in the permeation solution and recovered solution after permeation through NM-11 filters, most of the cell morphologies of the primary colon carcinoma (M-24) cells showed coagulated spherical cells in the permeation solution and recovered solution after permeation through PLGA-10/SK filters in this study. These results suggest that PLGA-10/SK filters are effective for the purification of colon carcinoma cells from primary colon carcinoma cells to remove fibroblast-like spindle-shaped cells. The residual rate, recovery rate, and permeation rate of the primary colon carcinoma (M-24) cells were evaluated after filtration through NM-11 and PLGA-10/SK filters, and the results are shown in Figure 7B. The permeation rate of primary colon carcinoma (M-24) cells through both NM-11 and PLGA-10/SK filters was found to be much less than that of HT-29 cells in this study (Figure 3B) and LoVo cells reported previously [13], whereas the recovery rate of primary colon carcinoma (M-24) cells through both the NM-11 and PLGA-10/SK filters was much higher than that of HT-29 cells in this study (Figure 3B) and LoVo cells reported previously [13]. In particular, the residual rate of primary colon carcinoma (M-24) cells from the PLGA-10/SK filters was much higher than that of HT-29 cells in this study (Figure 3B) and LoVo cells reported previously [13]. Primary colon carcinoma (M-24) cells were found to be more easily stacked on the filter pores of PLGA-10/SK filters than that of commercially available LoVo cells and HT-29 cells.

Because of the small number of primary colon carcinoma (M-24) cells derived from the patient’s colon tumor tissue, we could only evaluate CEA production by primary colon carcinoma (M-24) cells before and after permeation through PLGA-10/SK filters, and the results are shown in Figure 7C.

CEA production by primary colon carcinoma (M-24) cells in recovered solution after filtration through PLGA-10/SK filters was found to be higher than that of primary colon carcinoma (M-24) cells cultured on TCP dishes (control) and showed CEA production similar to that of LoVo cells and much higher than that of HT-29 cells. Primary colon carcinoma (M-24) cells in the permeation solution or migrated cells from PLGA-10/SK filters produced less CEA than primary colon carcinoma (M-24) cells cultured on TCP dishes (control) and cells in the recovered solution. Therefore, the filtration method using PLGA-10/SK filters can isolate colon carcinoma cells with characteristics of high CEA production from patient primary colon carcinoma cells in the recovered solution in this study.

## 4. Discussion

Both the colon cell line HT-29 and the patient’s primary colon carcinoma cells were filtered through porous NM-11 and PLGA/SK filters to enrich CSCs (CICs) in this study. Several cells, such as adipose-derived stem cells [16,29,30]; hematopoietic stem cells [26,27]; and different colon cell lines, e.g., LoVo cells [13], were purified using the filtration method. However, this study is the first to purify CSCs (CICs) from primary colon tumor cells derived from patient tumor tissue. The permeation rate of colon carcinoma cell lines, HT-29 in this study and LoVo cells in a previous study [13], was found to be relatively high, approximately 80%, by the filtration method through NM-11 and PLGA/SK filters, whereas the permeation rate of primary colon carcinoma (M-24) cells was only approximately 35%. This is probably because primary colon carcinoma cells contain several impurities, such extracellular matrices and fibroblasts, which contribute to the cells being plugged on the filter pores. Extremely high CEA production of the primary colon carcinoma (M-24) cells in the recovered solution through PLGA/SK-10 filters was found in this study, whereas high expression of CSC marker and high colony-forming ability of the commercially available colon carcinoma cells (HT-29 in this study and LoVo cells in the previous study [13]) were found in the migrated cells and the cells in the recovered solution. Therefore, primary colon carcinoma cells established in early passages (less than 10 passages in this study) seem to have some different characteristics from the commercially available colon carcinoma cell lines of HT-29 and LoVo cells. However, the filtration method was found to be effective for the purification of CSCs (CICs) for both the patient’s primary colon carcinoma cells and commercially available colon carcinoma cells.

Several cell purification methods have been established to collect cells with specific characteristics. Typical methods are (i) magnetic-activated cell sorting (MACS) [44,45,46], (ii) fluorescence-activated cell sorting (FACS) [47,48], and (iii) cell selection by genetic editing with antibiotic resistance genes and antibiotic treatment (e.g., puromycin, neomycin, G418, and zeocin) [49,50,51,52,53]. In both the MACS and FACS methods, antibody binding on the targeting cells is essential, which indicates the possibility of antibody contamination on the final product of the specific cells, and antibody contamination in the final product is not desirable for clinical usage of the specific cells. Furthermore, genetically edited cells are not appropriate for clinical usage. On the other hand, the filtration method that was developed by the authors’ group [26,27,28,29,30,31] does not need to use animal-derived antibodies or genetic editing, which is a safe and less laborious process of cell sorting to purify the targeted cells.

## 5. Conclusions

The filtration method was applied to enrich CSCs (CICs) from a colon carcinoma cell line (HT-29) and primary colon carcinoma cells derived from patient colon tumor tissue. The cells in the recovered solution and migrated cells through PLGA-5/SK and PLGA-10/SK filters showed higher CS marker expression, higher colony-forming ability, and higher CEA secretion than the cells cultivated on TCP plates using colon carcinoma HT-29 cells, which suggested that CSCs (CICs) would be purified from the cells in the recovered solution and migrated cells. High CEA secretion of primary colon carcinoma (M-24) cells was obtained in the recovered solution through PLGA/SK-10 filters, indicating that the purity of primary colon carcinoma cells can be enriched in the recovered solution by the filtration method through PLGA/SK-10 filters. This research successfully verified that the filtration method is effective for purification of primary colon carcinoma cells derived from patient colon tumor tissues as well as for colon carcinoma cell lines. However, primary colon carcinoma cells derived from only one patient’s colon tumor tissue were purified using the membrane filtration method in this study. Therefore, it should be necessary to verify our purification method of the membrane filtration method using primary colon carcinoma cells derived from multiple patient colon tumor tissues in future. It is also necessary to develop more efficient porous polymeric filters to purify CSCs (CICs) with higher purity in future.

## Figures and Tables

**Figure 1 polymers-13-03411-f001:**
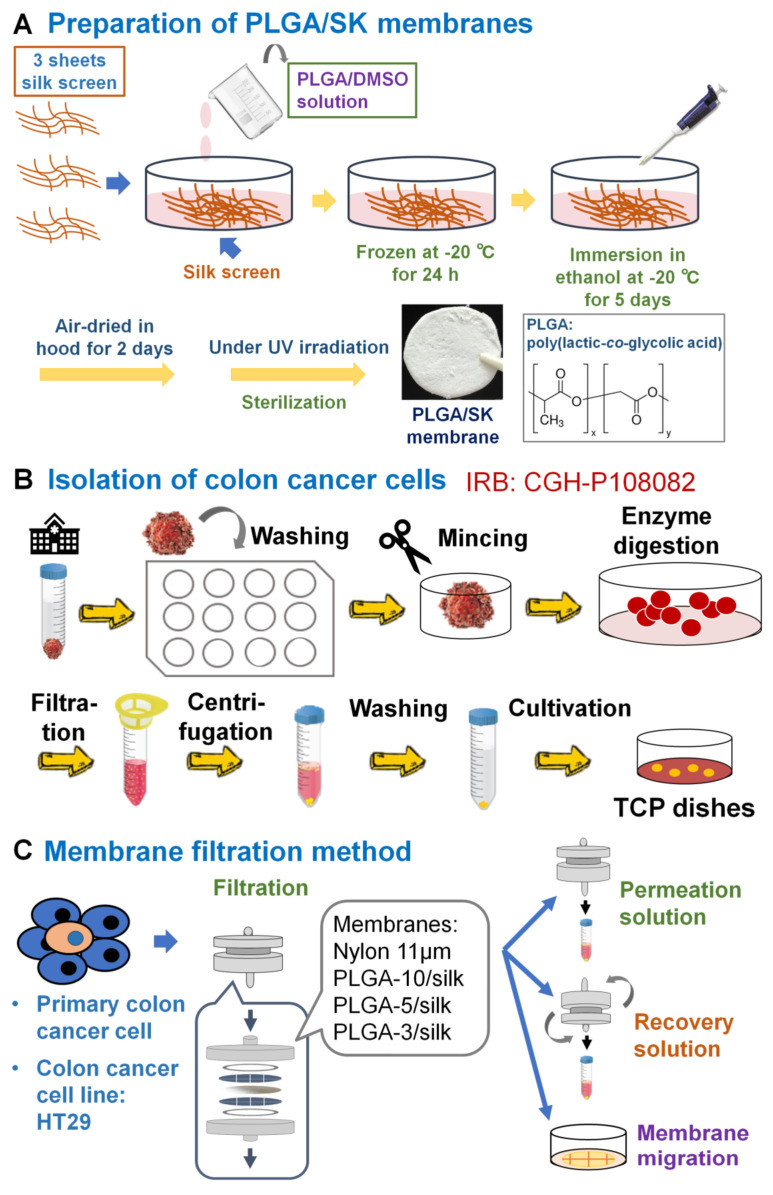
Isolation and filtration method of CSCs (CICs) from a colon carcinoma cell line and primary colon carcinoma cells derived from patient colon tumor tissue. (**A**) PLGA/SK filters prepared using the freeze-extraction process. (**B**) Isolation method of primary colon carcinoma cells from patient colon tumor tissues. Adenocarcinoma colon carcinoma tissues were minced into small sections and digested with collagenase IV. Subsequently, the primary colon carcinoma cell solution was seeded on TCP dishes and incubated in a 5% CO_2_ incubator to isolate primary colon carcinoma cells. (**C**) Filtration method to isolate CSCs (CICs) from colon carcinoma cells (HT-29) and primary colon tumor cells derived from patient colon tumor tissues. The cell solution was permeated via PLGA/SK or NM-11 filters to obtain permeation solution. Then, the cell culture media were permeated via the filters to obtain the recovered solution. Subsequently, the filters were transferred to TCP dishes containing cell culture medium and incubated for 6 days, after which the cells in the filters migrated into TCP dishes (the migrated cells). These cells were analyzed for CEA production, colony forming, and CSC (CIC) surface marker expression utilizing flow cytometry (BD Accuri™ C6, BD Biosciences, Franklin Lakes, NJ, USA).

**Figure 2 polymers-13-03411-f002:**
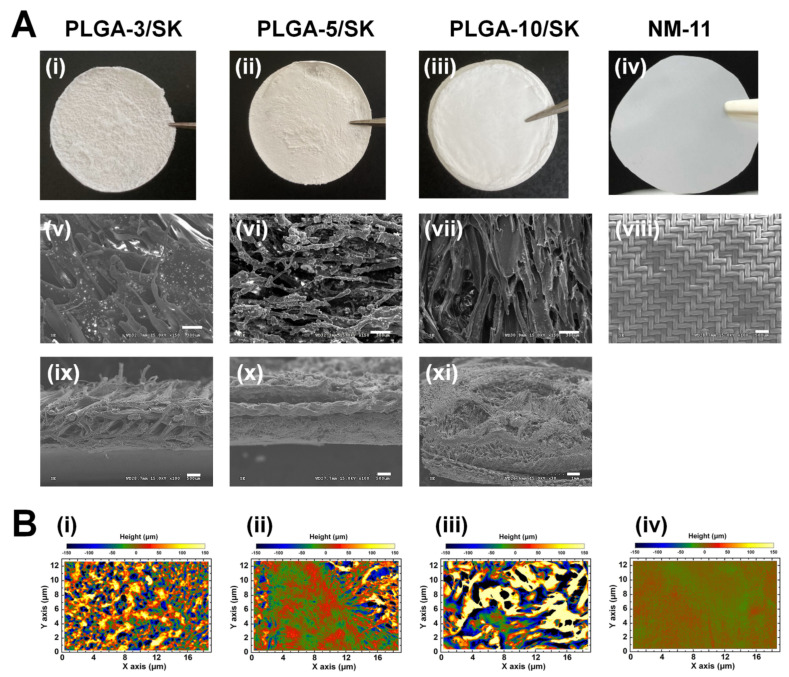
Filter configurations and morphologies used for isolation of CSCs (CICs) from colon carcinoma cells. (**A**) Outline photos of PLGA-3/SK (**i**), PLGA-5/SK (**ii**), PLGA-10/SK (**iii**), and NM-11 (**iv**) filters. Morphologies of PLGA-3/SK (**v**,**ix**), PLGA-5/SK (**vi**,**x**), PLGA-10/SK (**vii**,**xi**), and NM-11 (**viii**) filters from the top views (**v**–**viii**) and cross-sectional views (**ix**–**xi**) measured by SEM. The scale bar indicates 100 μm (**v**–**x**) and 300 μm (**xi**). (**B**) 3D surface image of PLGA-3/SK (**i**), PLGA-5/SK (**ii**), PLGA-10/SK (**iii**), and NY-11 (**iv**) filters analyzed using PRIMOS Lite.

**Figure 3 polymers-13-03411-f003:**
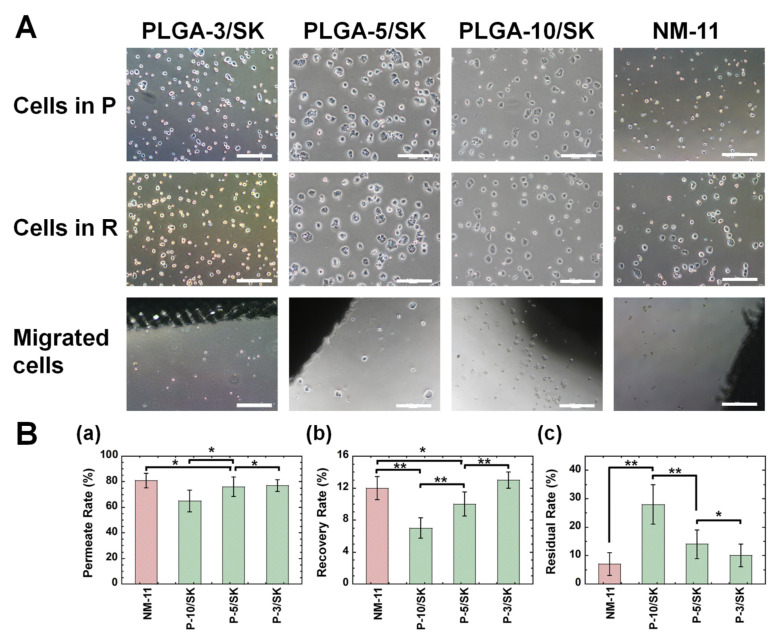
Morphologies and filtration characteristics of colon carcinoma cells after filtration from the filters. (**A**) Morphologies of colon carcinoma (HT-29) cells in permeation solution (P) (**i**–**iv**), recovered solution (R) (**v**–**viii**), and migrated cell (**ix**–**xii**) morphologies via the PLGA-3/SK (**i**,**v**,**ix**), PLGA-5/SK (**ii**,**vi**,**x**), PLGA-10/SK (**iii**,**vii**,**xi**), and NM-11 (**iv**,**viii**,**xii**) filters. The scale bar indicates 500 μm. (**B**) (**a**) Permeation rate of colon carcinoma (HT-29) cells via the NM-11, PLGA-10/SK (P-10/SK), PLGA-5/SK (P-5/SK), and PLGA-3/SK (P-3/SK) filters. (**b**) Recovery rate of colon carcinoma (HT-29) cells via the NM-11, PLGA-10/SK (P-10/SK), PLGA-5/SK (P-5/SK), and PLGA-3/SK (P-3/SK) filters. (**c**) Residual rate of colon carcinoma (HT-29) cells via the NM-11, PLGA-10/SK (P-10/SK), PLGA-5/SK (P-5/SK), and PLGA-3/SK (P-3/SK) filters. * *p* > 0.05, ** *p* < 0.05.

**Figure 4 polymers-13-03411-f004:**
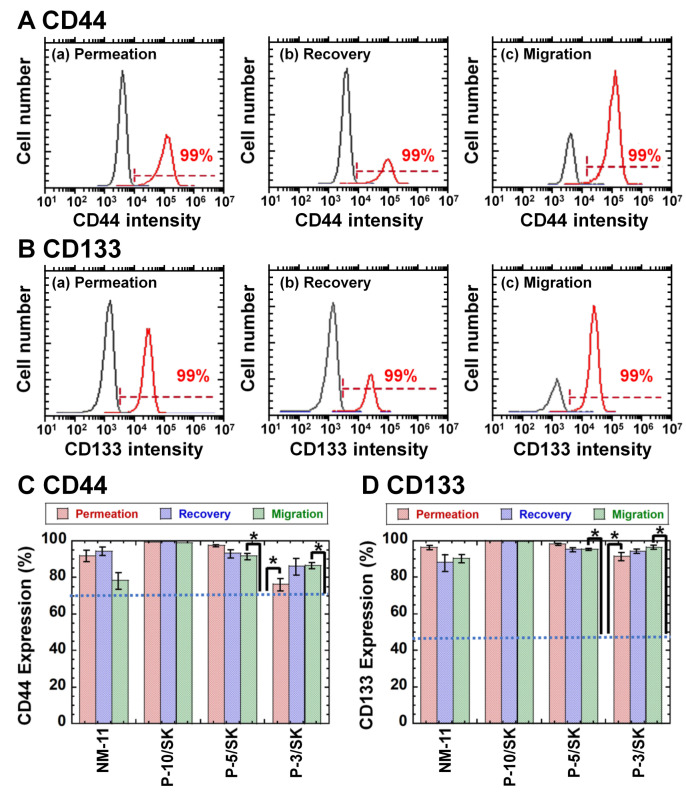
CD44 and CD 133 (CSC marker) expression in colon carcinoma HT-29 cells after filtration via NM-11 and PLGA/SK filters. (**A**) Flow cytometry chart of CD44 expression on HT-29 cells in the permeation solution (**a**), recovered solution (**b**), and migrated cells (**c**) via PLGA-10/SK filters. (**B**) Flow cytometry chart of CD133 expression on HT-29 cells in permeation solution (**a**), recovered solution (**b**), and migrated cells (**c**) via PLGA-10/SK filters. (**C**) CD44 expression on colon carcinoma HT-29 cells in the recovered solution, permeation solution, and cells migrated via NM-11, PLGA-10/SK (P-10/SK), PLGA-5/SK (P-5/SK), and PLGA-3/SK (P-3/SK) filters. (**D**) CD133 expression on colon carcinoma HT-29 cells in the recovered solution, permeation solution, and cells migrated via NM-11, PLGA-10/SK (P-10/SK), PLGA-5/SK (P-5/SK), and PLGA-3/SK (P-3/SK) filters. The dotted lines indicate CD44 (**C**) and CD133 (**D**) expression on colon carcinoma HT-29 cells incubated on TCP plates. * *p* < 0.05.

**Figure 5 polymers-13-03411-f005:**
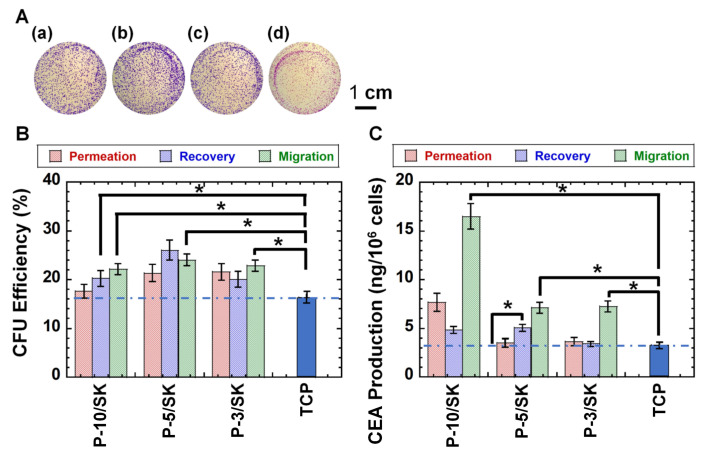
Colony-forming unit analysis and CEA production of colon carcinoma HT-29 cells after filtration via the NM-11 and PLGA/SK filters. (**A**) Colony-forming photos of colon carcinoma HT-29 cells in the permeation solution (**a**), recovered solution (**b**), and migrated cells (**c**) through the PLGA-10/SK filters. The colony-forming photo of the cells cultivated on TCP plates is also described (control) (**d**). The bar indicates 1 cm. (**B**) The number of colony-forming colon carcinoma HT-29 cells in the permeation solution and recovered solution and the cells that migrated through the PLGA-10/SK (P-10/SK), PLGA-5/SK (P-5/SK), and PLGA-3/SK (P-3/SK) filters as well as the number of colony-forming colon carcinoma HT-29 cells that were cultured on TCP dishes (control). (**C**) CEA production of colon carcinoma HT-29 cells in the permeation solution and recovered solution and the cells migrated through the PLGA-10/SK (P-10/SK), PLGA-5/SK (P-5/SK), and PLGA-3/SK (P-3/SK) filters as well as CEA production of colon carcinoma HT-29 cells, which were cultured on TCP dishes (control). * *p* < 0.05.

**Figure 6 polymers-13-03411-f006:**
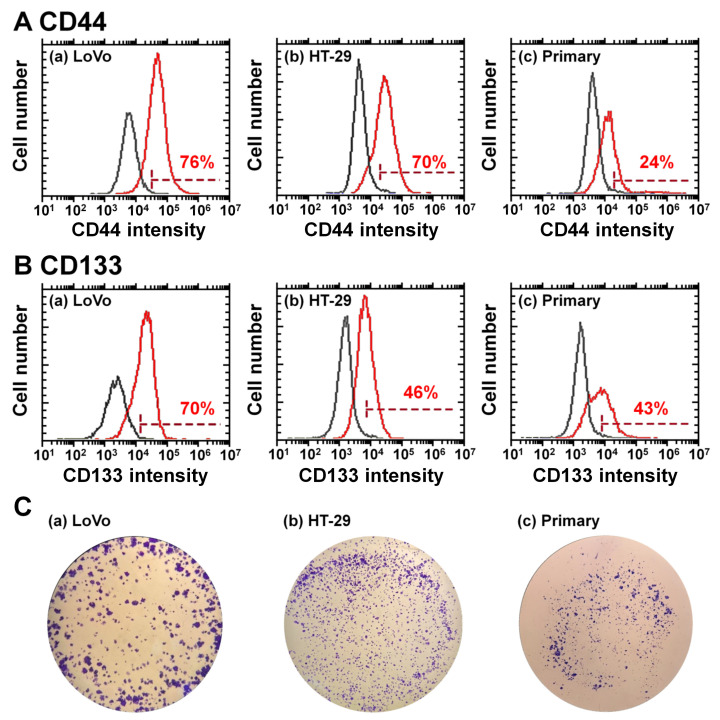
Characterization of primary colon carcinoma (M-24) cells derived from patient colon tumor tissues. (**A**) Flow cytometry chart of CD44 expression on LoVo cells (**a**), HT-29 cells (**b**), and primary colon carcinoma cells derived from patient colon tumor tissues (**c**). (**B**) Flow cytometry chart of CD133 expression on LoVo cells (**a**), HT-29 cells (**b**), and primary colon carcinoma cells (**c**). (**C**) Colony-forming photos of colon carcinoma LoVo cells (**a**), HT-29 cells (**b**), and primary colon carcinoma (M-24) cells derived from patient colon tumor tissues (**c**).

**Figure 7 polymers-13-03411-f007:**
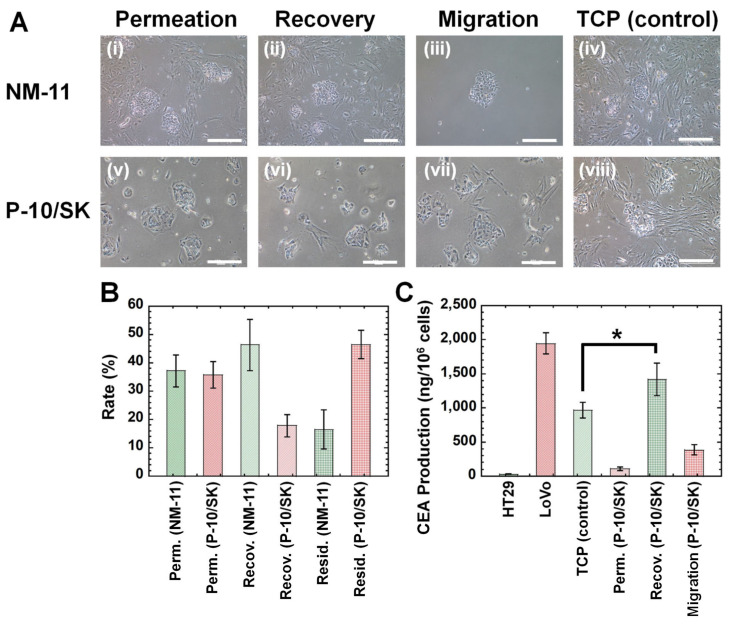
Morphologies, filtration characteristics, and CEA production of primary colon carcinoma (M-24) cells derived from patient colon tumor tissues after filtration via NM and PLGA/SK filters. (**A**) Morphologies of primary colon carcinoma (M-24) cells derived from the patient’s colon tumor tissues in permeation solution (**i**,**v**), recovered solution (**ii**,**vi**), and migrated cells (**iii**,**vii**) through the NM-11 (**i**–**iii**) and PLGA-10/SK (P-10/SK) (**v**–**vii**) filters. The morphologies of primary colon carcinoma (M-24) cells cultured on TCP dishes (control) are also shown (**iv**,**viii**). The scale bar indicates 100 μm. (**B**) Residual rate (Resid.), recovery rate (Recov.), and permeation rate (Perm.) of primary colon carcinoma (M-24) cells from the patient colon tumor tissues through NM-11 and PLGA-10/SK (P-10/SK) filters. (**C**) CEA production of primary colon carcinoma (M-24) cells derived from patient colon tumor tissues before (cell culture on TCP) and after permeation through PLGA-10/SK (P-10/SK) filters in permeation solution (Perm.), recovered solution (Recov.), and migrated cells (Migration). CEA production of HT-29 and LoVo cells cultured on TCP dishes is also shown (control). * *p* < 0.05.

**Table 1 polymers-13-03411-t001:** Materials used in this study.

Materials	Abbreviation	Catalog No.	Company
Polymer			
Poly(lactide-*co*-glycolic acid) (lactide:glycolic = 75:25)	PLGA	P1941	Sigma-Aldrich (St. Louis, MO, USA)
Silk screen	Silk	170 mesh	Yuzawaya, Tokyo, Japan
Nylon mesh filter (r = 11 µm)	NM11	NY1104700	Merck KGaA (Darmstadt, German)
Cells			
HT-29 cells	HT-29 cells	60157	BCRC, Food Industry Research and Development Institute (Hsinchu, Taiwan)
Cell culture dishes			
Six-well tissue culture polystyrene plate	TCPS	353046	Corning (Corning, NY, USA)
Chemicals			
2-Hydroxyethyl agarose	Agarose	A4018	Sigma-Aldrich (St. Louis, MO, USA)
High-vacuum grease	High-vacuum grease	1658832	Dow Corning Corporation, Midland, MI, USA
Human CEA ELISA kit	Human CEA ELISA kit	EHCEA	Thermo Fisher Scientific Inc. (Waltham, MA, USA)
Cell culture medium and component			
DMEM	DMEM	D5648-10×1 L	Sigma-Aldrich (St. Louis, MO, USA)
Fetal bovine serum	FBS	04-001-1A	Biological Industries, Kibbutz Beit-Haemek, Israel
Hoechst 33342	Hoechst	PA-3014	Lonza (Basel, Switzerland)
Surface markers			
7-AAD viability dye	7-AAD	559925	BD Biosciences (San Jose, CA, USA)
FITC mouse anti-human CD44	FITC anti-CD44	555478	BD Biosciences (San Jose, CA, USA)
PE mouse anti-human CD133/1	PE anti-CD133	130-080-801	Miltenyi Biotec (Bergisch Gladbach, North Rhine-Westphalia, Germany)
FITC mouse IgG2bκ, isotype control	FITC isotype	555742	BD Biosciences (San Jose, CA, USA)
PE mouse IgG1κ, isotype control	PE isotype	555749	BD Biosciences (San Jose, CA, USA)

**Table 2 polymers-13-03411-t002:** Characterization of membranes used for membrane filtration and migration methods.

	Membranes			
	NM-11	PLGA-10/SK	PLGA-5/SK	PLGA-3/SK
Average pore size (μm)	11.2 ± 2.1	28.9 ± 8.9	34.0 ± 8.2	36.3 ± 12.5

## Data Availability

Data are provided in the article.

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
