# Peer review of "Purification of Colon Carcinoma Cells from Primary Colon Tumor Using a Filtration Method via Porous Polymeric Filters"

_polymers, 2021, doi:10.3390/polym13193411_

Round 1

Reviewer 1 Report

The article of Jia-Hua Wang et al. with the title “Purification of colon carcinoma cells from primary colon tumor using a filtration method via porous polymeric filters” is an informative study aiming to highlight the way of analyzing the purification of CSCs (CICs) of colon carcinoma HT-29 cell line as well as primary colon carcinoma cells from colon cancer tissues of patients, by applying filtration method via NM and PLGA/SK filters. Overall, it is a well-presented and decent study. However, there are several issues that need to be addressed:

Major issues:

  • Results: It is recommended that the authors further justify of choosing HT-29 (either it is the morphological characteristics causing filtration or not).
  • Discussion: Due to the differentiations noted in the filtration methods, the authors should analyze more elaborately the differences between the commercially available colon carcinoma cell lines, LoVo and HT-29, and the primary colon carcinoma cells, M-24.
  • Conclusions: In this section, the authors should include weaknesses of the study and examine further cases to support their hypothesis. Additionally, they could clarify their future research perspectives in order to connect in a better way the results of their study with the related research of the field.

Minor issues:

  • Introduction: The authors should provide additional information on colorectal cancer.
  • The language of the manuscript could be improved.

Reviewer 2 Report

In this paper, the authors used a filtration method to purify colon carcinoma cells. The topic is interesting and the research well conducted but, before publication, some improvements are needed.

Some sentences are difficult to read. See, for example, line 51.

In line 60, the authors introduced the LoVo cell line without explaining what it is. Considering that Polymers is not a journal dedicated to cells or cancers, some readers could not know that they are epithelial human cells. Few words have to be inserted when they are named for the first time.

In lines 59-73, the literature regarding this kind of purification is limited by a paper published by the authors (ref 13). There are no other papers on the subject?

Round 2

Reviewer 1 Report

The issues raised by the Reviewer were properly addressed.